

# LLaVA-ReID: Selective Multi-image Questioner
# for Interactive Person Re-Identification

Yiding Lu [1]  Mouxing Yang [1]  Dezhong Peng [1]  Peng Hu [1]  Yijie Lin[* 1]  Xi Peng[* 1 2]

https://github.com/XLearning-SCU/LLaVA-ReID

## Abstract

Traditional text-based person ReID assumes that person descriptions from witnesses are complete and provided at once. However, in real-world scenarios, such descriptions are often partial or vague. To address this limitation, we introduce a new task called interactive person re-identification (Inter-ReID). Inter-ReID is a dialogue-based retrieval task that iteratively refines initial descriptions through ongoing interactions with the witnesses. To facilitate the study of this new task, we construct a dialogue dataset that incorporates multiple types of questions by decomposing fine-grained attributes of individuals. We further propose LLaVA-ReID, a question model that generates targeted questions based on visual and textual contexts to elicit additional details about the target person. Leveraging a looking-forward strategy, we prioritize the most informative questions as supervision during training. Experimental results on both Inter-ReID and text-based ReID benchmarks demonstrate that LLaVA-ReID significantly outperforms baselines.

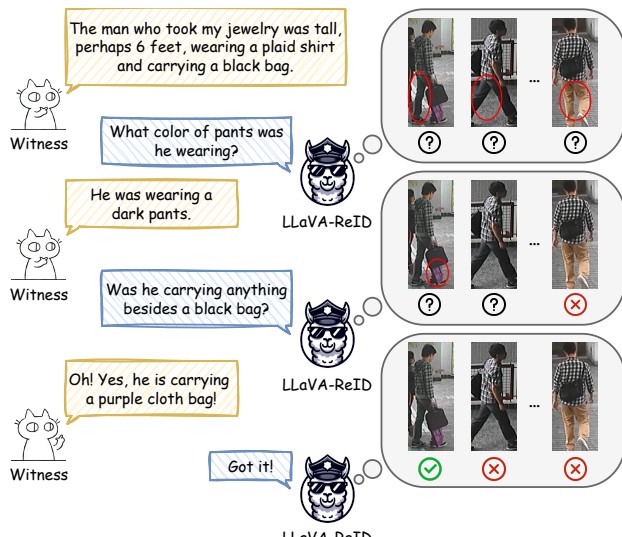

Figure 1. An illustrated example of interactive person re-identification. The red circles highlight the distinctive details in the candidate images that the inquiry process needs to focus on.

## 1. Introduction

*Never trust to general impressions, my boy, but concentrate yourself upon details.*

*Sherlock Holmes*

Imagine Sherlock Holmes standing in a dimly lit room, questioning a witness to a recent crime. The witness, nervous but determined, recounts the few fleeting moments he observed the suspect: "The man was tall, perhaps 6 feet, wearing a plaid shirt and carrying a black bag", as depicted in Figure 1. For Holmes, ever the skeptic, vague impressions are never enough and every detail matters. He investigates further: "What color of pants was he wearing? Was he carrying anything besides a black bag? Did he limp or walk with a peculiar gait?" Much like an artist refining a blurry sketch, Holmes sharpens the picture of the suspect through targeted inquiries about details, gradually uncovering the full truth.

Consider Holmes is equipped with an interactive tool, a system designed to help him ask increasingly refined and precise questions based on the witness's evolving description and the candidate suspects assessed from these clues. Rather than passively accepting general descriptions, this tool empowers Holmes to analyze both the description and the candidate suspects, dynamically adapting questions in real time. By guiding the witness to recall specific, crucial details about the suspect's appearance or behavior, the sys-

[1]College of Computer Science, Sichuan University, China [2]National Key Laboratory of Fundamental Algorithms and Models for Engineering Numerical Simulation, Sichuan University, China. [*]Correspondence to: Yijie Lin <linyijie.gm@gmail.com>, Xi Peng <pengx.gm@gmail.com>.

*Proceedings of the $42^{st}$ International Conference on Machine Learning*, Vancouver, Canada. PMLR 267, 2025. Copyright 2025 by the author(s).

tem continuously refines the portrayal of the suspect with each response, enabling Holmes to pinpoint the criminal more effectively.

In this paper, we introduce interactive person re-identification (Inter-ReID), a framework that learns to do the same: iteratively refines an initial description through ongoing interaction with the witness. To the best of our knowledge, no prior work has explored this new problem and the closest paradigm is text-based person ReID (T-ReID). However, T-ReID (Ye et al., 2021; Ding et al., 2021; Zuo et al., 2024) is fundamentally different, as it assumes a static, single-shot description provided in isolation. In contrast, Inter-ReID closely mirrors real-world scenarios, where initial descriptions are often partial and vague, requiring interactive refinement to accurately identify the target person.

To facilitate the study of this novel task, we construct a new dataset, *Interactive-PEDES*, which incorporates: i) coarse-grained descriptions to simulate the initial, partial queries provided by witnesses, ii) fine-grained descriptions that capture rich, detailed visual characteristics, simulating the witness's latent memories, and iii) multi-round dialogues derived by decomposing fine-grained descriptions into diverse questions, addressing detailed attributes of individuals.

In addition to the dataset, we address the challenge of generating specific and targeted questions to improve accuracy and adaptability, focusing on two core aspects: i) Representative Candidate Selection: we design a hard-pass selection model that identifies a subset of representative candidates from the gallery, emphasizing the critical differences between individuals. ii) Informative Question Generation: leveraging the distinct attribute-based questions provided in *Interactive-PEDES*, we propose a looking-forward strategy to dynamically select the most informative questions as supervision based on their potential information gain in each round of interaction. The contributions of this paper are as follows:

- We introduce a new task, Interactive Person Re-Identification, which goes beyond traditional T-ReID by incorporating interaction with witnesses to improve both accuracy and adaptability. To support this task, we construct a tailored dataset featuring multi-round dialogues, enabling more effective training and evaluation of Inter-ReID systems.

- We propose LLaVA-ReID, a multi-image question generator capable of identifying fine-grained differences between collections of images, leveraging both visual and textual contexts from candidate persons and dialogue history. Comprehensive experiments demonstrate that our method not only enhances performance in Inter-ReID but also benefits existing T-ReID tasks.

## 2. Related Work

### 2.1. Text-to-Image Person Re-identification

Person re-identification has gained significant attention as a critical task in surveillance and security systems (Ye et al., 2021), focusing on matching individuals across non-overlapping camera views (Yang et al., 2022) and varying modalities (Ye et al., 2020; Gong et al., 2024; Shi et al., 2024). Among the diverse Re-ID challenges, text-to-image ReID aims to retrieve the corresponding image of a person based on textual descriptions. The primary challenge lies in effectively aligning fine-grained visual features with language descriptions in a joint embedding space. Existing methods can be broadly categorized into two approaches: network architecture design and training objective design. The first category (Li et al., 2017; Gao et al., 2021; Shao et al., 2022) focuses on developing fine-grained fusion networks such as multi-scale interaction (Yan et al., 2023) mechanisms to encourage the alignment of visual details. The second category introduces auxiliary tasks to improve the fine-grained alignment between body regions and textual entities, such as random masking (Jiang & Ye, 2023; Bai et al., 2023), color prediction (Wu et al., 2021; Gong et al., 2022), and attribute prediction (Zuo et al., 2023).

These approaches typically assume that descriptions are complete and well-structured. However, in real-world scenarios, witnesses rarely provide a detailed and comprehensive account of the target person's appearance, often resulting in partial and vague descriptions. To address this limitation, we propose an interactive ReID framework that incorporates multiple rounds of dialogue between the witness and the retrieval system. This iterative process enables the system to refine the initial description by actively engaging the witness with targeted questions. Through active questioning, our system uncovers missing details, thus improving retrieval performance. While some recent works (Tan et al., 2024; Zuo et al., 2024; 2023) attempt to annotate more fine-grained descriptions, they focus on improving fine-grained alignment rather than guiding the witness to supplement overlooked details through a dialogue-based approach.

### 2.2. Interactive Cross-Modal Retrieval

Interactive cross-modal retrieval involves refining the initial query by engaging the user to provide additional information or clarification. Several studies have introduced interactive mechanisms in text-to-image retrieval (Levy et al., 2023; Lee et al., 2024; Zhu et al., 2024) and text-to-video retrieval (Madasu et al., 2022; Liang & Albanie, 2023), investigating various interaction formats (Kovashka et al., 2015; Cai et al., 2021; Lee et al., 2021; Ding et al., 2025). Among these formats, question-answering using free-form text dialogue has shown the most promise, as it closely mirrors the natural way humans interact (Madasu et al., 2022).

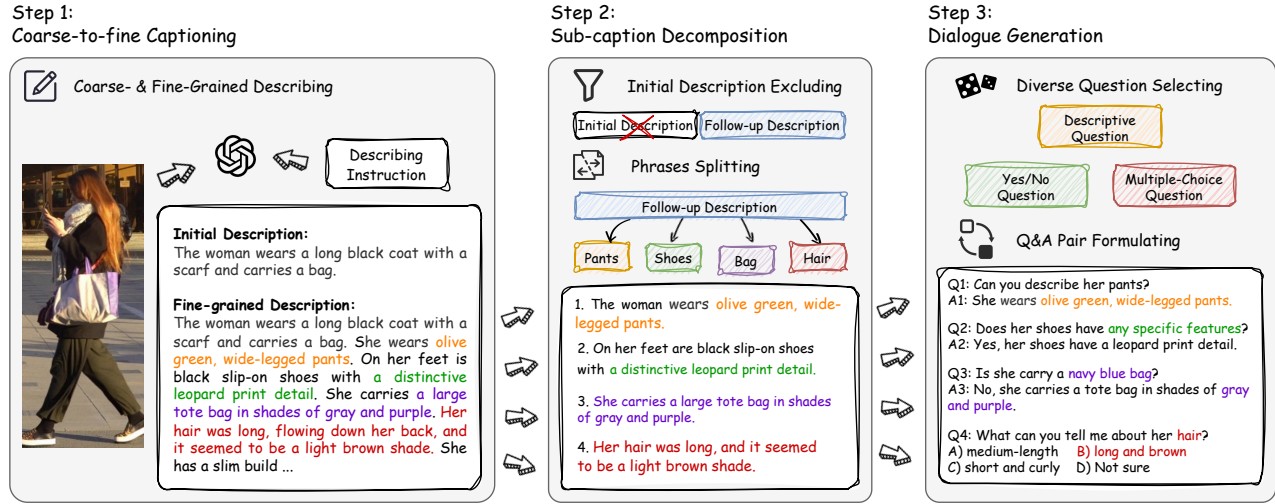

**Figure 2.** Illustration of our automated dialogue data construction pipeline. Step 1: Generate coarse and fine-grained descriptions. Step 2: Decompose follow-up descriptions into distinct attributes. Step 3: Formulate diverse Q&A pairs.

For example, SimIRV (Liang & Albanie, 2023) designs a heuristic question generator that asks about the attributes of objects in the image. Inspired by the context-aware capabilities of large language models (LLMs), ChatIR (Levy et al., 2023) leverages LLMs to generate the next question based on few-shot examples. More recently, PlugIR (Lee et al., 2024) proposes further enhancing performance by filtering redundant questions and rewriting the user query.

However, these methods primarily focus on general application scenarios and lack the domain-specific knowledge required for fine-grained retrieval tasks (Lin et al., 2023). To address this, we introduce the first customized dataset for training and evaluating Inter-ReID. The dataset includes subtle clothing features, accessories, and other distinctive attributes, enabling the questioner to effectively handle the nuances between individuals. Furthermore, we propose a question generator based on the large multi-modal model (Li et al., 2024b), which integrates both visual and textual contexts from candidate images and dialogue inputs. In contrast, existing methods first transform images into textual descriptions using image caption models (Li et al., 2022), which might lack the fine-grained visual details and multi-modal perception required for context-aware question generation.

## 3. Interactive-PEDES

In this section, we introduce our interactive person re-identification dataset, *Interactive-PEDES*. We first define the interactive person re-identification task (Section 3.1) and then describe the creation of the dataset (Section 3.2).

### 3.1. Interactive Person Re-Identification

Interactive person re-identification is a multi-round dialogue and retrieval process where an interactive system collaborates with a witness to identify the target person $I_{gt}$ from an image gallery $\mathcal{G} = \{I_1, I_2, \ldots, I_m\}$ using dialogue. The process begins with the witness providing an initial textual description $T$ of the target person. However, this initial description is often incomplete or vague, necessitating further clarification. To address this, the system employs iterative dialogue: in each round $t$, it generates a question $Q_t$ to guide the witness in recalling additional details about the target person. The witness responds to $Q_t$ with an answer $A_t$, and the resulting question-answer pair is incorporated into the dialogue context $\mathcal{D}_t = \{T, (Q_1, A_1), \ldots, (Q_t, A_t)\}$, which is used to identify the target person.

By leveraging the accumulated dialogue context, the system continuously refines its understanding of the target person, improving both retrieval accuracy and the generation of more precise, discriminative questions. This interactive cycle repeats until the target person is identified or a predefined maximum number of rounds is reached.

### 3.2. Dataset Construction

To simulate real-world scenarios where witnesses provide incremental and useful feedback, *Interactive-PEDES* includes both person images and corresponding dialogue for retrieval tasks. The dataset comprises 54,749 images of 13,051 individuals, collected from CUHK-PEDES (Li et al., 2017) and ICFG-PEDES (Ding et al., 2021). Each image is annotated with a dialogue that includes an initial description and an interactive dialogue detailing the person. The construction of the dataset involves three key steps: coarse-to-fine caption-

ing, sub-caption decomposition, and dialogue generation, as illustrated in Figure 2.

**Step 1: Coarse-to-Fine Captioning**  To generate dialogues in Inter-ReID, it is essential to define both the general impression and also more specific finer attributes that could recalled by the witness. For this purpose, we propose a coarse-to-fine captioning strategy utilizing GPT-4o (Hurst et al., 2024). Specifically, we first generate coarse-grained descriptions (*i.e.*, initial description $T$) that primarily focus on the overall appearance of the person, using the instructions randomly sampled from Table 5. Coarse-grained descriptions are typically brief and concise, simulating the initial high-level descriptions provided by the witness. Next, we generate fine-grained descriptions $F$ based on the initial descriptions, using the detailed prompt in Table 6. Fine-grained descriptions delve deeper into the details of the person, including attributes such as distinctive logos, accessories, hairstyle, walking posture, and even the environment or surroundings in which the witness encounters the person. Fine-grained descriptions are designed to simulate the latent memories of a witness, capturing distinguishing features that contribute to identifying the target person.

**Step 2: Sub-Caption Decomposition**  This step divides fine-grained descriptions into sub-captions to facilitate the construction of question-answer pairs in Step 3. Two key principles guide the creation of sub-captions: i) they must address specific details not mentioned in the initial description, and ii) each sub-caption should focus on a single and distinct aspect of the person. To achieve this, we first remove the information already provided in the initial description to generate a follow-up description that emphasizes the unexplored details. Next, we decompose the follow-up description into non-overlapping sub-captions using the prompt in Table 7. Each sub-caption represents a unique aspect of the person's features and ensures that no redundant questions will be asked.

**Step 3: Dialogue Generation**  To simulate real-world inquiry scenarios, we design diverse fine-grained questions, categorized into three types: descriptive questions (50%), yes/no questions (40%), and multiple-choice questions (10%). The questions are carefully crafted to extract detailed information from the sub-captions generated in Step 2, ensuring both diversity and depth in the dialogue. The corresponding prompt for generating each question $Q$ and its answer $A$ is provided in Table 8 and the order of Q&A pairs within the dialogue $\mathcal{D}_t$ is randomly chosen.

**Descriptive Questions** encourage the witness to describe details in their own words, promoting a richer and more natural response. For example, a question like *"Can you tell me about the style of the young man's coat?"* allows the

witness to provide specific information.

**Yes/No Questions** present an assumption about the target person, prompting the witness to confirm or deny the presence of a particular feature. As Agatha Christie wrote in *Dumb Witness*, *"I have often had occasion to notice how, where a direct question would fail to elicit a response, a false assumption brings instant information in the form of a contradiction."* Even when the assumption is incorrect, such questions can help stimulate the witness's memory about specific aspects of the target person. To simulate real-world scenarios, we design questions that could elicit either "yes" or "no" answers, ensuring a balanced and realistic mix of both correct and incorrect assumptions.

**Multiple-choice Questions** provide the witness with a set of options to choose from. The alternatives are carefully crafted to include subtle differences between features, such as dark brown vs. deep chestnut color. This format simplifies the witness's decision-making process while emphasizing specific and nuanced distinctions.

Finally, we construct the *Interactive-PEDES* dataset with an average of 9 dialogue rounds per image. The training set comprises 47,376 images corresponding to 11,543 identities, while the test set includes 7,373 images representing 1,508 identities. Additional details are provided in Appendix A.

## 4. LLaVA-ReID

In this section, we introduce our interactive person re-identification framework in Section 4.1 and provide a detailed explanation of our selective multi-image questioner (LLaVA-ReID) in Section 4.2.

### 4.1. Interactive ReID Framework

The interactive person re-identification framework consists of three main components: a Retriever, a Questioner (LLaVA-ReID, acting as Holmes's assistant), and an Answerer. Retriever identifies the target person by retrieving the person who best matches the descriptions from the image gallery. The Questioner generates discriminative and context-aware questions to elicit additional details about the target person. The Answerer simulates the role of the witness by responding to the questions with relevant information. This framework leverages iterative and interactive refinement of the description to enhance the precision of person re-identification.

**Retriever** is a dual-stream network (Radford et al., 2021) that encodes the dialogue descriptions and person images into a shared cross-modal space using a pre-trained visual encoder $\phi_v$ and textual encoder $\phi_t$. Specifically, the visual encoder $\phi_v$ first encodes the image gallery $\mathcal{G} = \{I_1, I_2, \ldots, I_m\}$ into embeddings $\{f_1, f_2, \ldots, f_m\}$

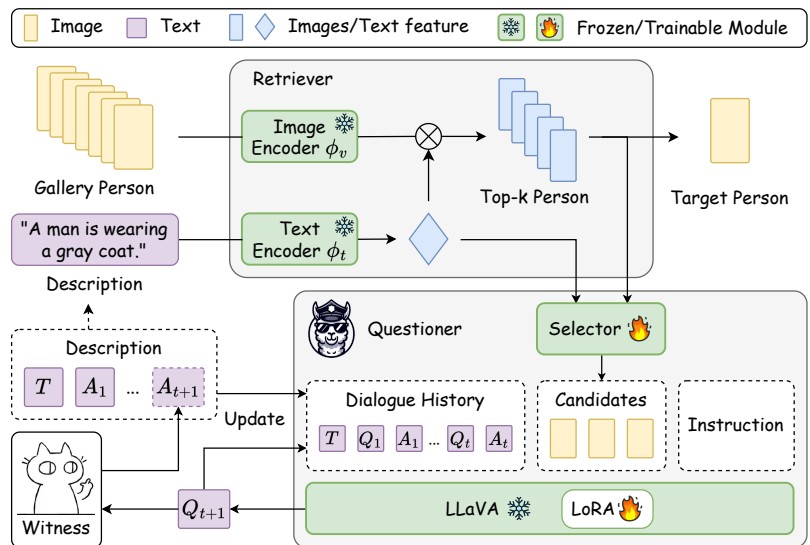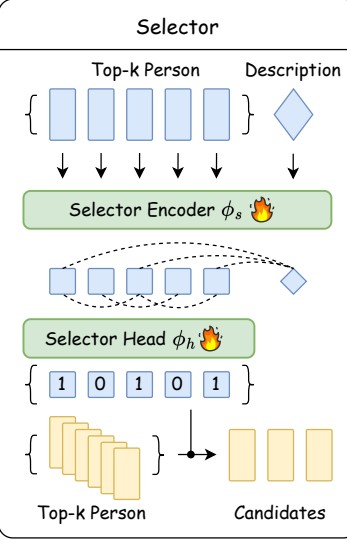

Figure 3. (*Left*) The framework of interactive person re-identification. The Retriever encodes gallery images and the description, providing retrieval results and the relevant candidates to the Questioner. The Questioner generates discriminative questions based on the description and the candidates. The Witness provides the corresponding information in response to these questions. (*Right*) The architecture of the selector. The selector chooses the most representative candidates from the top-$k$ person based on textual information.

in advance. The textual encoder $\phi_t$ encodes the context descriptions at the $t$-th round into an embedding $z_t = \phi_t(\{T, A_1, \ldots, A_t\})$. The matching probability of the $i$-th person is computed as:

$$p(I_i|\mathcal{D}_t) = \frac{\exp \text{sim}(z_t, f_i)}{\sum_j^m \exp \text{sim}(z_t, f_j)}, \quad (1)$$

where $\text{sim}(\cdot, \cdot)$ denotes the cosine similarity. The retrieval result $I^*$ is the image with the highest similarity, defined as $I^* = \arg\max_i p(I_i|\mathcal{D}_t)$.

**Questioner** is a large multimodal model (LMM) that generates specific questions $Q_t$ based on the currently collected information, including the dialogue $\mathcal{D}_{t-1}$ and the candidate set $\mathcal{C}_{t-1}$ retrieved from the person gallery. The candidate set should include representative individuals and encourage Questioner to effectively generate refined and contextually relevant questions.

**Answerer** is a large language model (LLM) that simulates the role of the witness. It receives questions and provides answers based on its memory (*i.e.*, the fine-grained descriptions $F$). An LLM is employed instead of an LMM since, in real-world scenarios, witnesses rely on recollections $F$ rather than directly viewing images of the target person. This aligns with the LLM's strength in processing and generating detailed text-based information. The prompt of Answerer is placed in Table 10.

### 4.2. Selective Multi-image Questioner

In this work, we develop a Questioner capable of generating increasingly refined and discriminative questions by leveraging both visual and textual contexts. Specifically, given the candidate set and the dialogue history from round $t-1$, the Questioner generates the question $Q_t$ with the maximum likelihood which is given by:

$$p(Q_t|\mathcal{C}_{t-1}, \mathcal{D}_{t-1}) = \prod_i^N p(q_i|q_{1:i-1}, \mathcal{C}_{t-1}, \mathcal{D}_{t-1}), \quad (2)$$

where $q_i$ is the $i$-th token of the generated question $Q_t$. To improve efficiency and relevance, the Questioner processes a concatenated input comprising candidate individuals, dialogue history, and instruction for question generation:

$$\texttt{[Images][Dialogue][Instruction]}, \quad (3)$$

where the instruction is randomly sampled from Table 9.

Notably, effective question generation involves addressing two fundamental challenges:

i) Image Selection: Which images should be chosen as context to help the Questioner identify specific attributes and generate targeted questions?

ii) Question Selection: How to determine the optimal sequence of questions from the dataset to maximize information gain and uncover critical details overlooked in previous dialogue rounds?

To address these challenges, we propose LLaVA-ReID, a selective multi-image Questioner that integrates Selective Visual Context and Looking-Forward Supervision to effectively tackle both challenges.

### 4.2.1. SELECTIVE VISUAL CONTEXT

The differences between candidate persons offer valuable insights for distinguishing the target person and are essential for enabling the Questioner to generate effective and discriminative questions. However, existing interactive retrieval methods often neglect this aspect, typically relying on top-$k$ selection (Liang & Albanie, 2023; Madasu et al., 2022) or $k$-means clustering (Lee et al., 2024) of candidates. Specifically, top-$k$ selection can limit the diversity of uncertain candidates in the gallery, reducing the range of potential questions the Questioner can generate while $k$-means clustering may fail to preserve fine-grained distinctions between candidates. Additionally, existing large multi-modal models struggle to process extensive image collections due to limited context understanding and computational constraints (Wu et al., 2024). Applying token reduction methods might damage the fine-grained features.

To address the above challenges, we propose a multi-image Questioner built on the LMM framework LLaVA (Li et al., 2024b), incorporating a novel hard-pass selection model preceding its vision encoder. The selection model identifies a subset of representative candidates from the gallery based on their relevance to the dialogue context, enabling the Questioner to focus on the most informative visual inputs. The structure of our selector is illustrated in Figure 3.

Specifically, we first retrieve the top $k$ most relevant person from the gallery given the current dialogue context $D_t$ using the Retriever model,

$$\mathcal{T} = \{I_i \mid i \in \text{top-}k(\text{sim}(z_t, f_i))\}. \quad (4)$$

Next, we feed the image embeddings and the dialogue embeddings into a shallow Transformer encoder $\phi_s$, enabling deep interaction between visual and textual modalities:

$$\mathbf{v} = \phi_s(f_c; z_t), \quad (5)$$

where $f_c = \{f_i \mid I_i \in \mathcal{T}\}$ denotes the embeddings of the candidate set $\mathcal{T}$. Here, $\mathbf{v}$ represents the conditional selection embeddings of each candidate person based on the textual dialogue. We treat the candidates as an unordered set and do not apply positional embeddings.

We then pass $\mathbf{v}$ through a linear layer $\phi_h$ to predict the final selection weight for each candidate person, *i.e.*, $\mathbf{w} = \text{Softmax}(\phi_h(\mathbf{v}))$. The final candidates sent to the LMM are the top-$c$ candidates with the highest weights:

$$\mathcal{C} = \{I_i \mid i \in \text{top-}c(\mathbf{w})\}. \quad (6)$$

To enable the selector to explore more possible combinations during the training stage, we employ a differentiable random sample strategy, namely, Gumbel-top-$k$ relaxations (Xie & Ermon, 2019). Specifically, given $k$ persons with weights $\mathbf{w}$, the probabilities of sampling a sequence $S_{\text{ordered}}$ of $c$ candidates without replacement are defined as follows:

$$p(S_{\text{ordered}} \mid \mathbf{w}) = \frac{\mathbf{w}_{i_1}}{Z} \frac{\mathbf{w}_{i_2}}{Z - \mathbf{w}_{i_1}} \cdots \frac{\mathbf{w}_{i_c}}{Z - \sum_{j=1}^{c-1} \mathbf{w}_{i_j}}, \quad (7)$$

where $Z = \sum_i^k \mathbf{w}_i$ is the normalizing constant and the subscript $i_1$ denotes the first selected candidate in the subset $\mathcal{C}$. The expected probability of sampling $\mathcal{C}$ is the sum over all permutations of candidates in $\mathcal{C}$:

$$\mathcal{C} \sim p(\mathcal{C} \mid \mathbf{w}) = \sum_{S_{\text{ordered}} \in \Pi(\mathcal{C})} p(S_{\text{ordered}} \mid \mathbf{w}), \quad (8)$$

where $\Pi(\cdot)$ denotes the set of all permutations. To build a differentiable subset, we perturb the weights $\mathbf{w}$ by adding Gumbel noise, *i.e.*, $\mathbf{w}_i = \mathbf{w}_i + g_i$ where $g_i \sim \text{Gumbel}(0, 1)$, and apply the re-parameterization trick to ensure the gradient flow is preserved throughout the sampling process. The selection model is supervised using the loss function derived from the looking-forward supervision introduced in the following section.

### 4.2.2. SELECTIVE SUPERVISION BY LOOKING-FORWARD

Although the constructed Q&A pairs in our interactive dataset *Interactive-PEDES* can serve as supervision, they are inherently an unordered set. Different questions yield varying levels of information gain depending on the current retrieval results. For example, if most candidate person are wearing dark pants, asking about the color of the pants might provide little discriminatory power, even though pants are an important attribute for identification. Conversely, if the candidates differ significantly in terms of accessories, asking about accessories could significantly narrow down the candidates. Thus, determining the optimal sequence of questions is critical to maximizing information gain.

A straightforward approach to this problem is to enumerate all possible permutations of question sequences and evaluate their impact exhaustively. However, the complexity of this brute-force strategy grows factorially with the number of rounds, *i.e.*, $O(t!)$, making it computationally infeasible. To address this, we propose a one-step looking-forward strategy that dynamically evaluates the information gain of each candidate question in the current round. This approach selects the most informative question based on its impact on the retrieval rank of the target person, avoiding the combinatorial explosion of full permutations.

Specifically, let $\mathcal{S}$ be the questions prepared in our dataset, and $\mathcal{Q}_{pre}^{t-1}$ be the questions asked in the previous $t-1$ rounds,

Table 1. Comparison to state-of-the-art interactive retrieval methods on Interactive-PEDES. "Initial" denotes using the initial description without interaction. Our method is marked in gray . A lower BRI indicates better performance.

| Method | Round 3 | | | | Round 5 | | | | BRI ↓ |
|---|---|---|---|---|---|---|---|---|---|
| | R@1 | R@5 | R@10 | mAP | R@1 | R@5 | R@10 | mAP | |
| Initial | 35.86 | 55.17 | 64.57 | 32.80 | 35.86 | 55.17 | 64.57 | 32.80 | - |
| SimIRV (Liang & Albanie, 2023) | 50.45 | 73.39 | 82.49 | 41.00 | 61.27 | 82.00 | 88.36 | 46.53 | 1.024 |
| ChatIR (Levy et al., 2023) | 57.85 | 78.69 | 86.08 | 44.92 | 63.86 | 83.81 | 89.84 | 48.05 | 0.935 |
| PlugIR (Lee et al., 2024) | 60.34 | 81.34 | 87.89 | 47.28 | 65.44 | 85.33 | 91.01 | 49.89 | 0.849 |
| LLaVA-ReID (Ours) | 63.96 | 84.70 | 91.27 | 48.45 | 73.20 | 90.62 | 95.95 | 53.31 | 0.719 |

with $\mathcal{Q}_{pre}^0 = \emptyset$ initially. At round $t$, we iterate through all candidate questions and append the corresponding answer to the description. We evaluate the impact of each question based on the retrieval rank of the target person. The question that yields the highest retrieval rank is selected as the supervision for the current round. Formally, the optimal question is determined as:

$$Q_t^* = \underset{Q_i \in (\mathcal{S} \setminus \mathcal{Q}_{pre}^{t-1})}{\arg\max} \ \mathrm{rank}\left(I_{gt}, \{T, A_1, \ldots, A_{t-1}, A_t^*\}\right),$$
(9)

where $\mathrm{rank}$ represents the retrieval rank of the target person $I_{gt}$ based on the context with the looking-forward description. The Questioner is then optimized using the negative log-likelihood (NLL) loss:

$$\mathcal{L}_{\mathrm{NLL}} = -\log p\left(Q_t^* \mid \mathcal{C}_{t-1}, \mathcal{D}_{t-1}\right).$$
(10)

This strategy ensures that the supervision dynamically adapts to retrieval results, prioritizing questions that provide the most information gain.

## 5. Experiments

In this section, we first introduce the details of the implementation of our approach. We then evaluate its performance on Inter-ReID, comparing it with interactive cross-modal retrieval methods. Next, we integrate our method into existing T-ReID frameworks to demonstrate its transferability in enhancing standard text-based retrieval. Finally, we conduct analysis studies and present qualitative results for further evaluation.

### 5.1. Implementation Details

For the Retriever, we adopt CLIP (Radford et al., 2021) as the backbone and train it using the IRRA (Jiang & Ye, 2023) framework with fine-grained descriptions from *Interactive-PEDES*. For the Questioner, we build our model on LLaVA-OneVision-Qwen2-7B-ov (Li et al., 2024a) and fine-tune it using QLoRA (Dettmers et al., 2023). For the Answerer, we use Qwen2.5-7B-Instruct (Team, 2024) to emulate witness

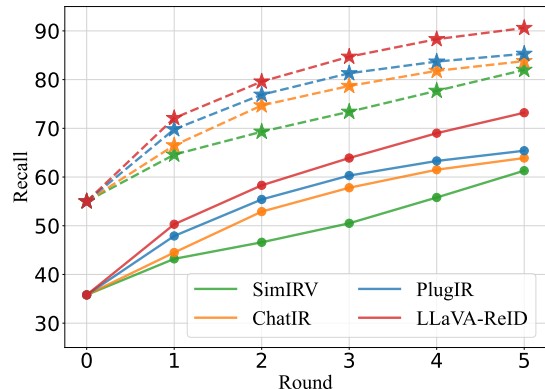

Figure 4. Retrieval performance v.s. Number of queries. The solid line denotes the R@1 and the dashed line denotes the R@5.

responses. We select 4 candidate images using our selector and set the maximum number of interaction rounds to 5. Module structures, training details, and evaluation metrics are provided in Appendix C.

### 5.2. Interactive Person Re-Identification

We compared our method against three state-of-the-art interactive cross-modal retrieval baselines: i) SimIRV (Liang & Albanie, 2023) is a heuristic method that extracts objects from the initial description and then generates detailed questions about these objects. ii) ChatIR (Levy et al., 2023) instructs pre-trained LLMs using multiple dialogue examples to guide the model to predict the next question. iii) PlugIR (Lee et al., 2024) selects candidates using $k$-means clustering and generates the next question based on the textual descriptions of these images. A filtering process iteratively removes redundant questions until a suitable one is selected, making this approach computationally inefficient.

As shown in Table 1, LLaVA-ReID achieves superior performance compared to baselines. In particular, it improves R@1 by 28.1% and 37.34% after 3 and 5 rounds of interaction (compared to Initial), respectively. Notably, LLaVA-ReID significantly outperforms the recent state-of-the-art

*Table 2.* Performance comparison of integration with existing T-ReID methods on three benchmarks.

| Method | CUHK-PEDES | | | | ICFG-PEDES | | | | RSTPReid | | | |
|---|---|---|---|---|---|---|---|---|---|---|---|---|
| | R@1 | R@5 | R@10 | mAP | R@1 | R@5 | R@10 | mAP | R@1 | R@5 | R@10 | mAP |
| CFine (Yan et al., 2023) | 69.57 | 85.93 | 91.15 | - | 60.83 | 76.55 | 82.42 | - | 50.55 | 72.50 | 81.60 | - |
| RaSa (Bai et al., 2023) | 76.51 | 76.51 | 94.25 | 69.38 | 65.28 | 80.40 | 85.12 | 41.29 | 66.90 | 86.50 | 91.35 | 52.31 |
| APTM (Yang et al., 2023) | 76.53 | 90.04 | 94.15 | 66.91 | 68.51 | 82.99 | 87.56 | 41.22 | 67.50 | 85.70 | 91.45 | 52.56 |
| AUL (Li et al., 2024c) | 77.23 | 90.43 | 94.41 | - | 69.16 | 83.32 | 88.37 | - | 71.65 | 87.55 | 92.05 | - |
| IRRA (Jiang & Ye, 2023) | 73.38 | 89.93 | 93.71 | 66.13 | 63.46 | 80.25 | 85.82 | 38.06 | 60.20 | 81.30 | 88.20 | 47.17 |
| + LLaVA-ReID | 78.51 | 92.43 | 95.67 | 70.61 | 67.44 | 82.91 | 87.69 | 40.60 | 69.85 | 88.10 | 92.55 | 54.92 |
| RDE (Qin et al., 2024) | 75.94 | 90.14 | 94.12 | 67.56 | 67.68 | 82.47 | 87.36 | 40.06 | 65.35 | 83.95 | 89.90 | 50.88 |
| + LLaVA-ReID | 79.39 | 93.02 | 95.96 | 71.70 | 69.98 | 84.32 | 88.60 | 42.12 | 72.00 | 88.95 | 93.55 | 56.06 |

method PlugIR, achieving improvements of 7.76% in R@1. Moreover, our method attains the lowest BRI (Lee et al., 2024), which measures the interaction efficiency and ranking improvement. This demonstrates the effectiveness of our method in refining retrieval results with fewer interactions. To further analyze retrieval performance across interaction rounds, we place the visualization results in Figure 4. While all methods benefit from iterative questioning, our approach exhibits the most significant performance gains, highlighting its ability to refine retrieval through more effective and targeted inquiries.

## 5.3. Integration with Text-based ReID

We further integrate LLaVA-ReID into existing T-ReID frameworks and evaluate its transferability on T-ReID benchmarks, including CUHK-PEDES (Li et al., 2017), ICFG-PEDES (Ding et al., 2021) and RSTPReid (Zhu et al., 2021). Specifically, we use the dataset annotations as the initial descriptions and conduct a 5-round interaction to refine the retrieval process. After the interaction, we encode the initial description using the existing T-ReID model, process the dialogue with our Retriever, and re-rank the matching scores by averaging the similarity. In this setup, T-ReID models capture the static initial descriptions, while our approach extracts additional details through interactive refinement.

As shown in Table 2, integrating our method into IRRA (Jiang & Ye, 2023) and RDE (Qin et al., 2024), where 5-round interaction consistently enhances performance across different methods and benchmarks, achieving an average R@1 improvement of 5.19%. Notably, our significant improvements on ICFG-PEDES, a dataset with human-annotated fine-grained descriptions, reveal that partial and vague descriptions remain inevitable, even when annotators are instructed to provide as much detail as possible. This highlights the necessity of interactive refinement to achieve more accurate and reliable ReID in real-world scenarios.

*Table 3.* Analysis of image and question selection strategy.

| Settings | | R@1 | R@5 | R@10 | BRI ↓ |
|---|---|---|---|---|---|
| Candidate | k-means | 72.51 | 90.08 | 94.59 | 0.727 |
| | uniform | 72.70 | 90.06 | 94.60 | 0.735 |
| | top-k | 72.53 | 89.78 | 94.22 | 0.740 |
| | selector | 73.20 | 90.62 | 95.95 | 0.719 |
| Question | random | 72.25 | 89.69 | 94.48 | 0.813 |
| | look-forward | 73.20 | 90.62 | 95.95 | 0.719 |

## 5.4. Analysis of Image and Question Selection

We conduct an analysis study to evaluate the impact of different selection strategies for candidate images and target questions. For candidate selection, we compare our proposed selector with top-$k$ (Liang & Albanie, 2023; Madasu et al., 2022), $k$-means (Lee et al., 2024), and uniform sampling. Top-$k$ selects the $k$ images with the highest similarity scores. $k$-means clusters the gallery and selects the closest sample from each centroid as representative candidates. Uniform sampling randomly selects images as candidates. For target question selection, we compare our looking-forward strategy with random sampling to assess the effectiveness of question selection in refining retrieval. As shown in Table 3, our selector outperforms the widely used strategies and highlights the benefits of selecting representative candidates. Meanwhile, our looking-forward strategy demonstrates its advantage in dynamically selecting informative questions.

## 5.5. Qualitative Results

We visualize several retrieval processes in Figures 7 and 8. As shown, LLaVA-ReID effectively focuses on fine-grained attributes such as shoes, hair, surroundings, and even makes assumptions about logos and accessories to refine the retrieval process. Specifically, LLaVA-ReID captures key distinguishing features among candidates, such as the dis-

tinctive logo on the jacket (round 4 in Figure 7) and the length of the shirt (round 3 in Figure 8), further validating the effectiveness of our selection. These examples demonstrate that LLaVA-ReID can effectively identify and emphasize the fine-grained attributes crucial for differentiating individuals in real-world scenarios.

## 6. Conclusion

This paper introduces Interactive Person Re-Identification, a new problem that more closely aligns with real-world scenarios by incorporating multi-round dialogue between the witness and the retrieval system. To support this task, we propose a specific dataset *Interactive-PEDES* and propose a multi-image questioner LLaVA-ReID that leverages both visual and textual contexts to identify fine-grained differences. Despite achieving strong performance, challenges remain, such as the limited ability of the retriever to handle incremental and dialogue-based descriptions. Future research could further explore these directions.

## Impact Statement

The proposed interactive person ReID framework has broad applications in public spaces such as airports, shopping centers, and transit hubs, aiding in locating missing children or elderly individuals. Additionally, this technology can assist law enforcement in identifying and tracking suspects, contributing to enhanced public security.

However, the deployment of such systems raises critical ethical concerns, particularly regarding privacy and surveillance. Ensuring responsible use requires stringent safeguards, including informed consent, data protection measures, and transparent governance.

## Acknowledgments

This work was supported in part by NSFC under Grant 62176171, 624B2099, 62472295, U24B20174; in part by the Fundamental Research Funds for the Central Universities under Grant CJ202303, CJ202403; in part by Sichuan Science and Technology Planning Project under Grant 24NSFTD0130; and in part by Baidu Scholarship.

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

## Appendix Overview

This supplementary document is organized as follows:

- Sec. A presents the details of our Inter-ReID dataset, *Interactive-PEDES*.

- Sec. B outlines the instructions for the Questioner and Answerer components in our interactive ReID framework.

- Sec. C provides the experimental details, including implementation specifics and evaluation metrics.

## A. *Interactive-PEDES* Details

In this section, we present additional details about our *Interactive-PEDES* dataset and provide the prompts used for its automatic generation.

### A.1. Details of the Dataset

In this section, we provide statistics and visualizations for our dataset and compare its description length and word diversity with other existing T-ReID datasets, as shown in Table 4. Specifically, Interact-PEDES-init and Interact-PEDES-fine refer to the initial and fine-grained descriptions in our dataset, respectively. Our dataset exhibits a higher level of text granularity, with an average length of 135 words, compared to other datasets, enabling more detailed and nuanced descriptions. The distribution of the description length and the dialogue round are shown in Figure 5. Our dataset provides comprehensive dialogue about the person's features.

Additionally, we present further visualizations in Figure 6. The left panel displays a word frequency distribution of the first six words in the generated questions, while the right panel illustrates a word cloud of the fine-grained descriptions, where the size of each word corresponds to its frequency in the dataset.

*Table 4.* Statistics of texts in our dataset compared to other existing T-ReID datasets.

| Dataset | Maximum Length | Minimum Length | Average Length | Unique Words |
|---|---|---|---|---|
| CUHK-PEDES | 112 | 13 | 26.6 | 7470 |
| ICFG-PEDES | 84 | 10 | 38.7 | 3102 |
| *Interact-PEDES*-init | 34 | 6 | 17.2 | 2394 |
| *Interact-PEDES*-fine | 193 | 75 | 135.4 | 6255 |

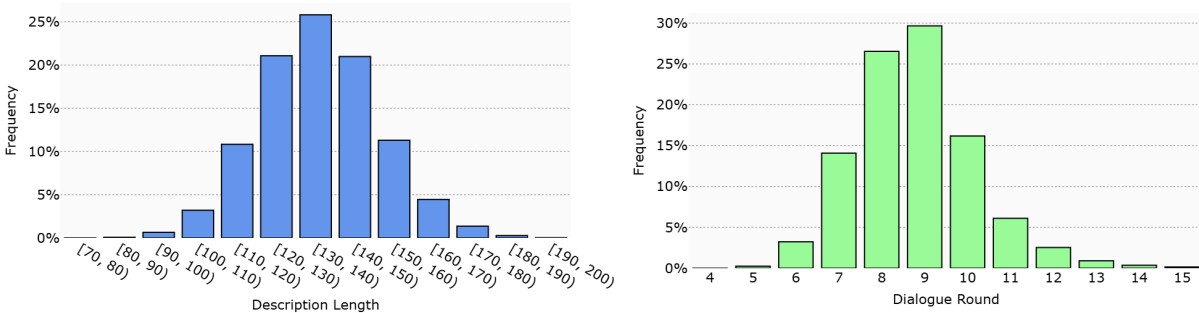

*Figure 5.* Distribution of description length (*Left*) and the dialogue round (*Right*) in our dataset.

### A.2. Prompts for Automatic Dataset Construction

In this section, we present the prompts used for the automatic construction of our dataset, covering the generation of coarse- and fine-grained descriptions, sub-caption decomposition, and interactive question-answer pairs, as detailed in Tables 5 to 8.

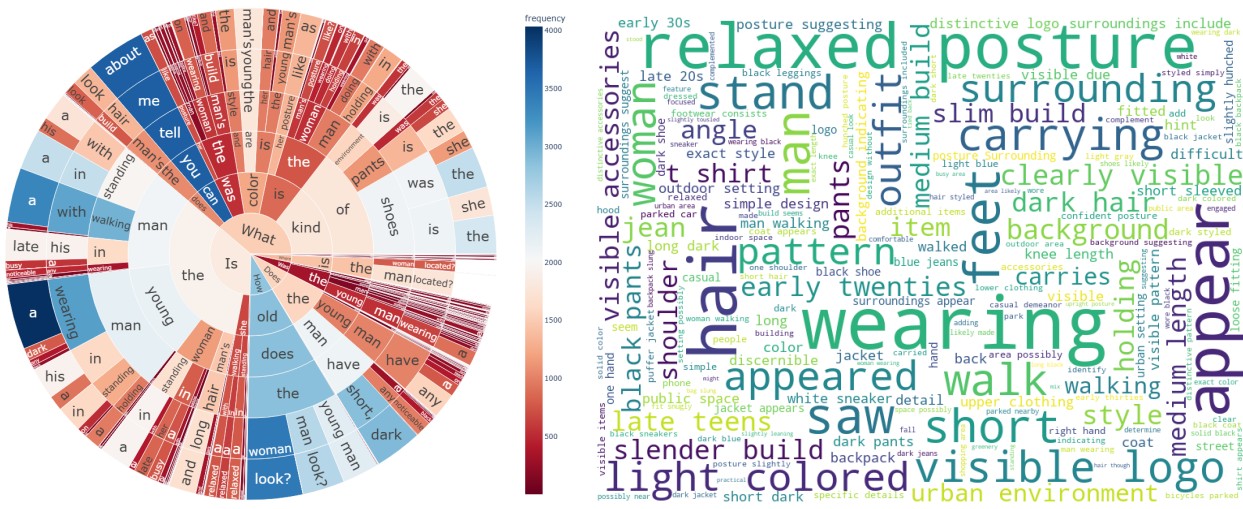

*Figure 6.* (*Left*) Word frequency statistics of question in our dataset. The word order of sentences starts at the center and extends outward. (*Right*) Word cloud of fine-grained descriptions. A larger area or font size indicates a higher frequency of occurrence.

*Table 5.* The list of instructions for coarse-grained description.

> **Prompts for Coarse-grained Description:**
>
> - Describe the person's clothing in one sentence.
> - Generate a brief caption about the clothing in one sentence.
> - Generate a brief caption of this person in one sentence. Begin with "I saw a".

*Table 6.* The instruction for fine-grained description. The `<init_query>` will be replaced by a generated coarse-grained description.

> **Prompt for Fine-grained Description:**
> Describe the person in the image using definitive statements. Focus on identifying and describing the following entities and attributes:
> - Approximate age range.
> - Color, type, patterns, and length of their upper and lower clothing, as well as any distinctive logos or details that stand out.
> - Color and type of shoes or other footwear.
> - Any items they are wearing, carrying, or holding, along with their colors, types, and patterns.
> - Color, length, and style of hair.
> - Build and posture.
> - Surroundings.
> Provide the most direct and lifelike description of this person in a paragraph, as if you are watching this person in the scenario. Focus on what is visible and what the person doesn't have that might aid in identifying them. Avoid using phrases like "The image", or "The picture". Use he, she, or descriptive terms (man, woman, boy, girl, lady) as subject rather than "The person".
> Begin with "`<init_query>`", and provide more detailed information.

*Table 7.* The instruction for sub-caption decomposition.

**Prompt for Follow-up Description Splitting:**
Given a person's description, summarize the content of the description into individual sentences. Each sentence should capture one aspect of the description, ensuring that no information is lost in the splitting process.
List the sentence as follows:
1. [first sentence]
2. [second sentence]
3. ...

Here is an example:
Description: A young woman, likely in her late teens to early twenties, is walking with a relaxed posture. She is wearing a dark blue puffer jacket that appears to be zippered with a logo on the back, providing warmth and comfort. Underneath, the specifics of her lower clothing are not visible, but she has on dark jeans that fit snugly. Her footwear consists of bright blue athletic shoes, which stand out against the darker tones of his clothing. She carries a large, dark-colored backpack, which seems practical for daily use. She had long, dark brown hair that fell past her shoulders, styled straight. The setting is indoors, possibly in a lobby or entrance area, with a glimpse of outdoor activity visible through the glass, indicating a busy environment.
Sentences:
1. A young woman is in his late teens to early twenties.
2. She is walking with a relaxed posture.
3. She is wearing a dark blue puffer jacket with a logo on the back, that appears to be zippered.
4. Underneath the jacket, the specifics of her lower clothing is not visible.
5. She has dark jeans that fit snugly.
6. Her footwear consists of bright blue athletic shoes.
7. She carries a large, dark-colored backpack.
8. She had long, dark brown hair that fell past her shoulders, styled straight.
9. The setting is indoors, possibly in a lobby or entrance area, with a glimpse of outdoor activity visible through the glass.

Now split the following description:
Description: <description>
Sentences:

*Table 8.* The instruction for dialogue generation, where `<subject>` will be replaced by "man" or "woman" based on the gender of the person and `<question_type_prompt>` is sampled from the question prompts below.

---

**Base Prompt for Q&A Pair Formulating:**
You are given a **[sentence]** describing a visible feature of an image of a `<subject>`. This may include details about the `<subject>`'s appearance, location, or surroundings. Your task is to generate a question about a feature of the `<subject>` described in the **[sentence]** (or details about the location or surroundings) that could help identify this person in a large collection of images.
The question should sound natural, as if you are asking a witness, without implying that you already know the answer. Avoid phrases like "in this image" or "How would you describe" and focus on describing visual features. Use appropriate pronouns (he, she) or descriptive terms (e.g., young man, elderly woman) when referring to the subject. Then, based on the **[sentence]**, provide a direct and complete answer.
`<question_type_prompt>`
**[sentence]**: `<sub_caption>`

**Question Prompts for Each Type:**

- **Yes/No question with positive answer:**
  The question should be a yes or no question. Format your response as follows:
  Question: **[question]**
  Answer: Yes, **[answer]**

- **Yes/No question with a negative answer:**
  The question should be a yes or no question that assumes something incorrect about the person. Then provide a corrective answer based on the original sentence. Format your response as follows:
  Question: **[question]**
  Answer: No, **[answer]**

- **Descriptive question:**
  The question should be a descriptive question that asks for the feature the sentence is describing. Format your response as follows:
  Question: **[question]**
  Answer: **[answer]**

- **Descriptive question (Wh- form):**
  The question should be a Wh- question seeking one specific piece of information that is described in the **[sentence]**. Format your response as follows:
  Question: **[question]**
  Answer: **[answer]**

- **Multiple-choice question:**
  The question should be a multiple-choice question with four options including:
  - the correct answer,
  - three possible options that perturb the type, length, and color. These altered options should not be easily distinguishable from the correct answer.
  The answer should be a complete sentence, rather than only an option. Format your response as follows:
  Question: **[question]**
  A) **[option A]**
  B) **[option B]**
  C) **[option C]**
  D) **[option D]**
  Answer: **[answer]**

---

# B. The instructions of Interactive ReID Framework

In this section, we provide the prompts used for the Questioner and Answerer, as detailed in Tables 9 and 10.

*Table 9.* The instruction for training LLaVA-ReID (Questioner).

**Instructions:**

- `<candidates><dialog>`
  Based on this information:
  **[Candidate Person]** The closest match to the provided description and answers.
  **[Similar Persons]** Individuals who also match but are less similar.
  **[Conversation]** The initial description from the witness and the ongoing dialogue.
  Generate a question that could help differentiate the candidate from similar individuals. Avoid repeating questions that have already been answered with "I don't know.".

- `<candidates><dialog>`
  Based on this information:
  **[Candidate Person]** The individual most likely matches the description provided.
  **[Similar Persons]** Individuals with similar characteristics, but they are less likely to be the candidate.
  **[Conversation]** The dialogue between the witness and investigator, including all available descriptions.
  Generate a question that explores a specific detail or feature that differentiates the candidate from others. Ensure the question has not already been answered or ruled out with an "I don't know." response.

- `<candidates><dialog>`
  The following context applies:
  **[Candidate Person]** The individual who best fits the description so far.
  **[Similar Persons]** Individuals who have overlapping characteristics but are less likely to be the target.
  **[Conversation]** The ongoing witness description and responses to questions.
  Generate a question that focuses on details that could clearly identify the candidate, while avoiding any redundant questions that have already been answered with "I don't know.".

**Image Tokens Placeholder:**
`<candidates>`= "[Candidate Person] `<images>`\n[Similar Persons]"
**for** i *in range(*`num_candidates-1`*)*:
$\quad$ `<candidates>`+=f"#{i} `<images>`\n"
**Dialog Context:**
`<dialog>`="[Conversation] Witness: `<init_query>`"
**for** `question, answer` *in* `previous_dialog`:
$\quad$ `<dialog>`+=f"You: {question}\n Witness: {answer}\n"

*Table 10.* The instruction for Answerer.

---

**Prompt for Answerer:**

Imagine you are a witness who has seen one person and someone is asking the information about that person. You are answering a question about that person and this is what you remember:

<context>

Answer the question only according to what you remember.

Instruction:

1. Answer directly in a complete sentence that provides the information to the question, without including any extra information.

2. Avoid giving answers that are just 'Yes', 'No', or a single word or phrase. Always answer in full sentences, even for multiple-choice questions.

3. If an entity or attribute is not included in your memory, assume the person does not possess it.

Question: <question>

Answer:

---

## C. Experimental Details

In this section, we provide additional details on the model structures, training configurations, and evaluation metrics used in our experiments.

### C.1. Model structures

**Retriever**. We use a pre-trained CLIP model (Radford et al., 2021), employing CLIP-ViT-B/16 as the visual encoder and CLIP-Xformer as the text encoder. The model is fine-tuned using the T-ReID method IRRA (Jiang & Ye, 2023) on the fine-grained descriptions from *Interactive-PEDES* for 30 epochs with a batch size of 128. All other training parameters follow the original IRRA settings. To handle longer text inputs, we interpolate the text positional embeddings from 77 to 192. Once trained, the Retriever is frozen for the remaining tasks.

**Questioner**. It is built on LLaVA-OneVision-Qwen2-7B-ov (Li et al., 2024a) and fine-tuned using QLoRA (Dettmers et al., 2023). The model is quantized to 4-bit and LoRA weights are applied with $r = 128$ and $\alpha = 256$. The learning rate is set to $1 \times 10^{-5}$, with a batch size of $4$ and gradient accumulation steps of $4$. To reduce the length of image patch tokens, $2 \times 2$ mean pooling is applied to each image, reducing the number of tokens to 49 per candidate image. The hyper-parameters for training Questioner are shown in Table 11, and the instruction used for question generation is provided in Table 8.

**Answerer**. We use Qwen2.5-7B-Instruct (Team, 2024) for emulating witness responses. The prompt used for the Answerer is provided in Table 10.

**Selector**. It is implemented using a 5-layer Transformer with a latent dimension of 512, followed by a linear layer. The number of candidate images is dynamically calculated as $k = \lceil \frac{200}{t} \rceil$, where $t$ is the current interaction round, with a fixed set of $c = 4$ candidates selected per round. Learnable modality embeddings are added to differentiate between visual and textual inputs, without using positional embeddings.

### C.2. Training Details

During training, we first warm up the Questioner and selector by training them separately with top-4 images as context using NLL loss and BCE loss for 1 and 20 epochs, respectively. After this, we combine the Questioner and the selector and train them jointly using NLL loss for 0.5 epoch. During the interaction process, we limit the question length to 96 tokens and the answer length to 40 tokens. All experiments are conducted on an Ubuntu 20.04 system with NVIDIA 4090 GPUs.

*Table 11.* Training configuration of LLaVA-ReID

| Parameter | Value |
|---|---|
| lora_alpha | 256 |
| lora_r | 128 |
| lora_dropout | 0.05 |
| deepspeed | zero2 |
| bf16 | True |
| epoch | 1 |
| batch size | 4 |
| learning rate | $1 \times 10^{-5}$ |
| weight decay | 0 |
| warm-up ratio | 0.02 |
| lv scheduler | cosine |
| model max length | 4096 |
| attn_implementation | sdpa |
| bits | 4 |
| double_quant | True |
| quant_type | nf4 |

## C.3. Evaluation Metrics

We evaluate interactive ReID performance using Recall@$k$, mAP, and BRI (Lee et al., 2024).

- Recall@$k$ measures the percentage of cases where the ground-truth person appears in the retrieved top-$k$ images ($k = 1, 5, 10$).
- mAP (mean Average Precision) provides an overall measure of retrieval accuracy by averaging precision across all queries.
- BRI (Best Log Rank Integral) assesses the system's ability to refine retrieval results over multiple interaction rounds. It measures the average area under the $\log$ rank curve of all rounds. A lower BRI value indicates a more efficient and effective refinement process.

To analyze the impact of interaction, we report Recall and mAP after 3 and 5 rounds in Table 1 to show the performance improvements over time.

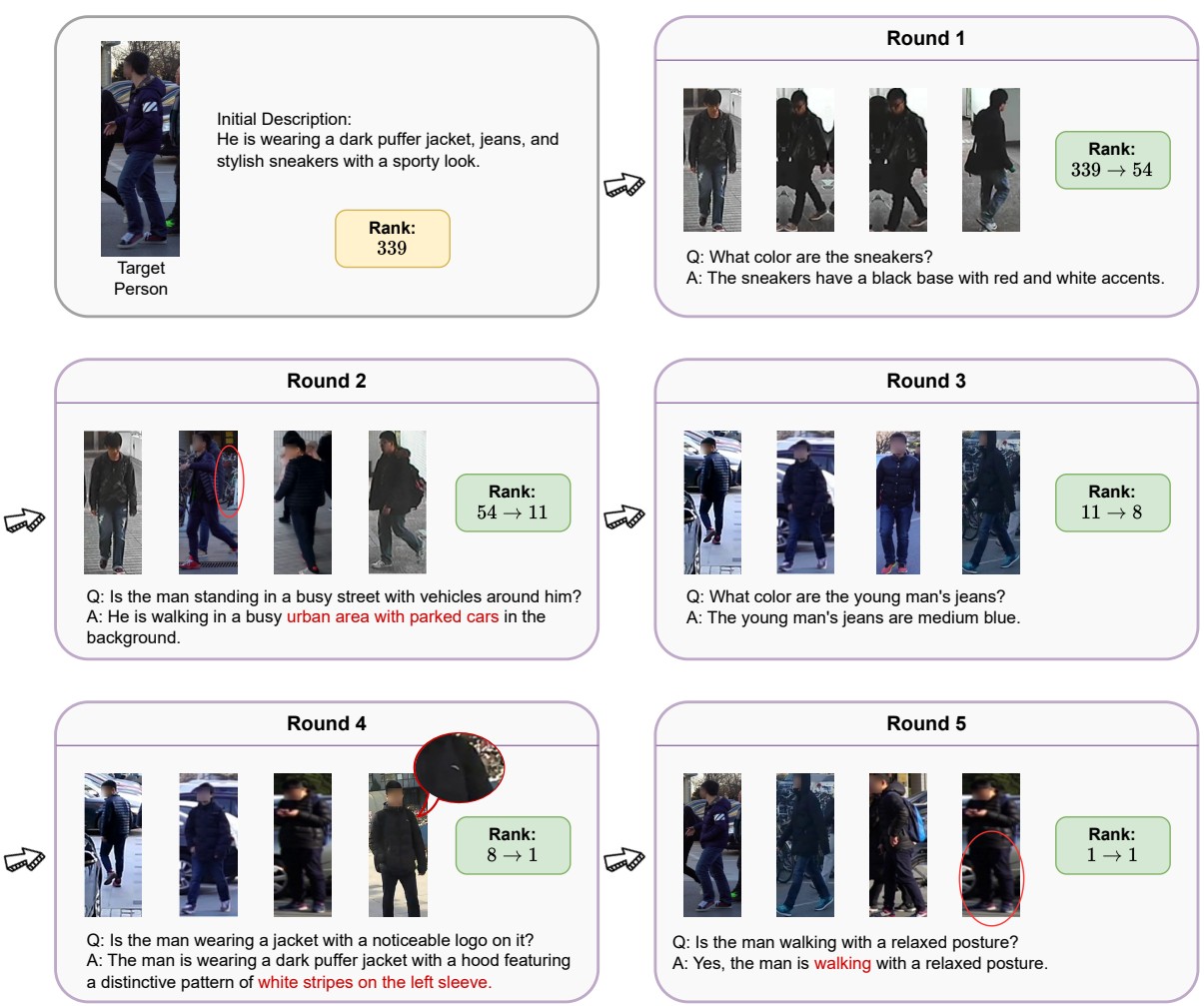

*Figure 7.* Qualitative results of the dialogue generated by our interactive system. The 4 images in each round are the representative candidates selected by our selector.

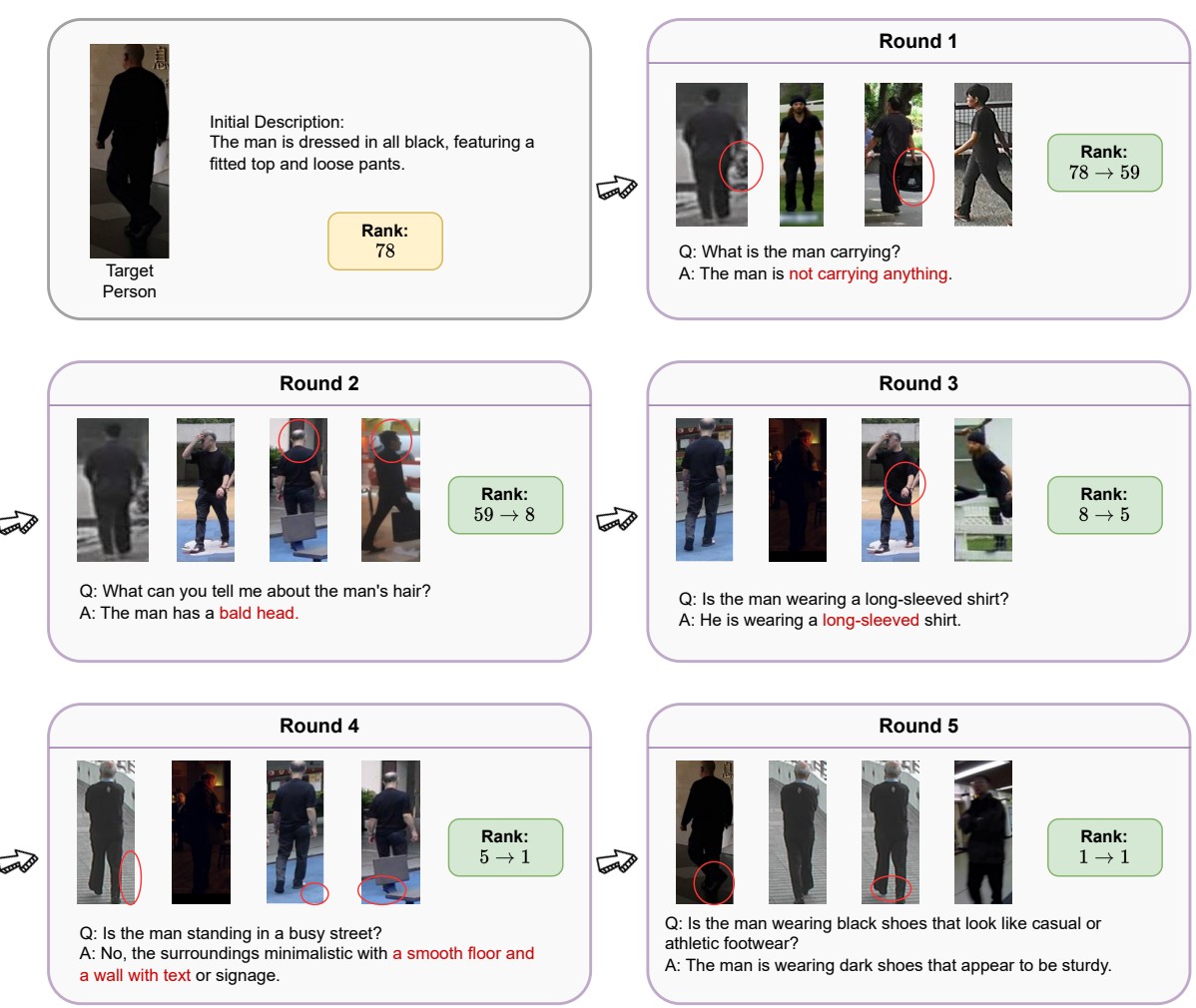

*Figure 8.* Qualitative results of the dialogue generated by our interactive system. The 4 images in each round are the representative candidates selected by our selector.

