# OpenReview forum: "LLaVA-ReID: Selective Multi-image Questioner for Interactive Person Re-Identification"
_ICML.cc/2025/Conference — ICML 2025 poster_

### Official Review · Reviewer_1zUH · 2025-03-05

**Overall Recommendation:** 4

**Summary:**

This paper introduces a new task named interactive person re-identification (I-ReID), which aims to address the insufficient details in initial human descriptions. To support this task, the paper contributes a new dataset comprising multi-round dialogues generated using a customized approach with vision-language models (VLMs). Another key contribution is a novel interactive framework tailored for the ReID domain. The well-designed question model effectively identifies the most informative questions from the gallery, significantly enhancing performance. Experimental results demonstrate the superiority of this task-specific interactive framework over existing methods from other domains and show its ability to supplement missing details in text-based ReID methods.

**Claims And Evidence:**

Yes. The main technical claims are: (i) **Representative Candidate Selection** emphasizes critical differences between individuals. (ii) **Informative Question Generation** dynamically selects the most informative questions for supervision. Both claims are well-supported by extensive experiments.

**Essential References Not Discussed:**

The authors discuss and compare a wide range of related methods.

**Experimental Designs Or Analyses:**

Yes. I have reviewed all experimental designs and results. Table 1 and Figure 1 present performance comparisons against existing interactive retrieval methods in other domains. Table 2 highlights the effectiveness of the proposed method over existing text-based ReID methods, underscoring the necessity of interactive ReID for supplementing missing details. Additionally, ablation studies confirm the indispensability of each module.

**Methods And Evaluation Criteria:**

Yes. The paper proposes a multi-round dialogue benchmark for the interactive person ReID task. This benchmark includes three types of Q&A pairs, which could facilitate further research in this area.

**Other Comments Or Suggestions:**

Please see the weaknesses.

**Other Strengths And Weaknesses:**

**Strengths:**
1) The paper introduces a novel and practical task: interactive person ReID.
2) It provides the first dialogue-based dataset for this task.
3) The proposed approach is novel, and experiments confirm its effectiveness over existing interactive methods from other domains.
4) The framework can be integrated into text-based ReID methods to mitigate the issue of insufficient human-annotated details, significantly improving performance.

**Major Weaknesses:**
1) GPT-like VLMs are prone to hallucinations. Have the authors considered this issue in dataset construction? Could leveraging existing dataset annotations help mitigate hallucinations?
2) Why does the instruction for generating fine-grained captions start with coarse-grained captions? Would an alternative approach better leverage the VLM’s captioning capabilities?
3) Figure 3 needs revision: (i) The input to the text encoder should not be limited to the initial description, and (ii) the meaning of "clue" in the figure is undefined.
4) Will the text input exceed the maximum input length of the CLIP model?
5) It is unclear how the question model functions when the interactive framework is integrated with existing text-based methods.

**Minor Weaknesses:**
1) Steps ii and iii in dataset construction appear somewhat redundant and could be merged for clarity.
2) The abbreviation "I-ReID" may be confused with the widely recognized image-based ReID. A different abbreviation is recommended.

**Questions For Authors:**

Please see the weaknesses.

**Relation To Broader Scientific Literature:**

The contribution of the paper could be summarized in two main aspects.
i) New task. Considering the practical demands of security and protection, the paper releases a novel task named interactive person reid, which involve progressively revealing the most relevant person according to the witness's description.
ii) Novel approach. The proposed method could not only emphasize the critical differences between individuals, but also dynamically select the most informative questions as supervision. Extensive experiments verify the effectiveness of the method. More importantly, the experiment results show off that the interactive framework could bring performance gain for the existing text-based reid methods, demonstrating the insufficient details in the existing human-annotated data.

**Theoretical Claims:**

The paper does not involve theoretical claims.

---

> ### Author Rebuttal · Authors · 2025-03-31
>
> Thanks a lot for reviewing our paper and giving us valuable suggestions. We will answer the questions one by one.
>
> > Q1: Visual hallucination in VLM captioning.
>
> Thanks for your insightful comments. We acknowledge that GPT-like vision-language models (VLMs) are prone to hallucinations, particularly when generating fine-grained descriptions. Leveraging existing techniques, such as contrastive decoding [1], in data construction could further improve the accuracy and reliability of our descriptions. We will explore these enhancements in future work.
>
> [1] Leng S, Zhang H, Chen G, et al. Mitigating object hallucinations in large vision-language models through visual contrastive decoding[C]//Proceedings of the IEEE/CVF Conference on Computer Vision and Pattern Recognition. 2024: 13872-13882.
>
> > Q2: Explanation of coarse-to-fine captioning.
>
> In interactive ReID, we expect the witness to provide additional fine-grained discriminative information in the subsequent dialogue rounds. This information should not merely repeat what has already been stated. Therefore, we use the coarse description as a basis for generating the follow-up description.
>
> An alternative approach could be to generate a fine-grained description first and then remove the overlapping content with the initial description. However, our generation strategy simplifies this process of decomposing the description, making it more efficient and structured.
>
> > Q3: Revision of figure 3
>
> (i) Thanks for your advice, we will change it to "Descriptions from Witness".
>
> (ii) The term "clue" refers to the embeddings of the context description $z_t$. We use this term to indicate that the candidates are selected based on the current "clue" we have at hand.
>
> We apologize for any confusion caused by this terminology and will revise the figure in the next version.
>
>
> > Q4: Long text encoding of CLIP model
>
> As mentioned in Appendix C, we interpolate the text positional embeddings from 77 to 192 and finetune fine-grained captions in Interactive-PEDES to obtain the Retriever. If the total text length exceeds 192 tokens, we truncate it accordingly. In practice, since the initial description has a maximum length of 48 tokens, the average length per turn in a five-round dialogue is (192-48)/5=28.8. This is sufficient for the current interaction setup, and we can also further interpolate it to longer encoding lengths.
>
>
> > Q5: Details about the question model functions when the interactive framework is integrated with existing text-based methods.
>
> Please refer to our response to Q1 from reviewer fi2Z.
>
>
> > Q6: Re-organize section 3.2 (dataset construction)
>
> Thanks for your valuable suggestions. We will revise this section to improve the clarity.
>
> > Q7: Abbreviation of interactive person re-identification.
>
> Thank you for pointing this out. We will update the abbreviation to "Inter-ReID" in the new version.

---

> > ### Comment · Reviewer_1zUH · 2025-04-02
> >
> > The author's reply has mostly cleared up my questions, so I'll stick with the original score.

---

### Official Review · Reviewer_Ty38 · 2025-03-08

**Overall Recommendation:** 4

**Summary:**

In this paper, a novel task is presented to address the limitations of traditional text-based ReID which rely on complete and one-time descriptions from witnesses. This work introduces interactive person re-identification (I-ReID) that employs multi-round question-answer dialogue to iteratively gather information of person. To achieve this, a question model is designed to generate context-aware questions based on both visual and textual cues. Additionally, the authors construct a dataset tailored to I-ReID and employs a looking-forward strategy to prioritize the most informative questions. Experimental results demonstrate its effectiveness.

**Claims And Evidence:**

The paper's central claim is that witnesses often struggle to provide complete and clear descriptions of a target person on their own. This claim is well-supported by researches in interactive cross-modal retrieval and is intuitively valid within the text-based ReID domain. Furthermore, experimental results validate that even datasets with human-annotated, fine-grained descriptions inevitably contain partial or ambiguous information.

**Essential References Not Discussed:**

No

**Experimental Designs Or Analyses:**

I have reviewed the experimental analyses, and the experimental design appears sound and well-structured.

**Methods And Evaluation Criteria:**

The proposed methods and evaluation criteria are appropriate for the problem setting. Recall and mAP are standard metrics in text-based person ReID, and BRI (introduced in recent interactive retrieval research) is also used to evaluate the effectiveness of the interactive refinement process. Overall, the proposed methodology and evaluation framework align well with the problem context and practical application.

**Other Comments Or Suggestions:**

1.	There is a typo in section 5.2 (We compared…)
2.	The author should add an interpretation of BRI (e.g. lower is better) in the caption of Table 1.

**Other Strengths And Weaknesses:**

### Strengths
1.	The interactive framework could be a new research direction for the community, improving the applicability of text-based ReID in real-world cases.
2.	The paper constructs a new dataset with initial descriptions, fine-grained descriptions, and multi-round dialogues, enabling more effective training and evaluation of interactive ReID methods.
3.	The proposed candidate selection model provides representative sample to the model and effectively benefit the question generation. Experimental results demonstrate the effectiveness of the proposed method.

### Weaknesses
1.	The multi-images questioner will lead to additional computational overhead. The authors should provide a comparison of inference time between their method and baseline models. Additionally, the number of candidate images is likely a critical factor in performance; an ablation study on this aspect would strengthen the analysis.
2.	The expression clarity of Gumbel top-k sampling should be improved. While the rationale behind using Gumbel top-k sampling is understandable, its role and the necessity of the Gumbel trick should be more clearly explained.

**Questions For Authors:**

1.	In section 5.3, what contextual information does the answer model use? Given that existing datasets lack additional fine-grained descriptions, how is this experiment conducted?
2.	The figure 7 seems to be confusing. If the rank is 1, why is the target person not visible among the given images?

**Relation To Broader Scientific Literature:**

This paper introduces an interactive setting for text-based person ReID. Previous studies mainly focus in improve the fine-grained matching capability and test on the static and complete description in existing dataset. However, real-world scenarios often involve incomplete or ambiguous descriptions provided by witnesses in isolation. This paper bridges that gap by proposing an interactive refinement approach, enhancing both applicability and robustness of text-based ReID in practical settings.

**Theoretical Claims:**

The paper does not propose new theoretical claims.

---

> ### Author Rebuttal · Authors · 2025-04-01
>
> We appreciate the insightful questions and will address your concerns in the following.
>
> > Q1: Comparison of inference time and ablation study on the number of candidates.
>
> We conduct these experiments on NVIDIA RTX 3090 GPUs. The inference time for the 5-round question generation of LLaVA-ReID and other baseline methods are shown in the table below:
>
> | Method         | Inference Time (second per person) | R@1 |
> | -------------- | ---------------------------------- | ----------------- |
> | SimIRV         | 0.007                              | 61.3              |
> | ChatIR         | 0.216                              | 63.9              |
> | PlugIR         | 13.06                              | 65.4              |
> | **LLaVA-ReID** | **0.621**                          | **73.2**          |
>
> Our multi-image questioner does not add significant computational overhead and achieves the best performance.
>
> The retrieval results for different numbers of candidate images are shown in the table below:
>
> | Number of Candidates | R@1       | R@5      | R@10     | mAP       | BRI       |
> | -------------------- | --------- | -------- | -------- | --------- | --------- |
> | 2                    | 72.4      | 89.7     | 94.5     | 53.2      | 0.738     |
> | **4 (ours)**         | **73.2** | **90.6** | **96.0** | **53.3** | **0.719** |
> | 6                    | 71.4      | 89.2     | 94.2     | 52.8      | 0.764     |
> | 10                   | 71.6      | 88.9     | 93.8     | 52.7      | 0.772     |
>
> The results indicate that setting $k=4$ provides the best balance between retrieval performance and computational efficiency.
>
> In our previous experiments, the reported BRI was computed using a logarithm with base 2. According to the recently released code of BRI[1], we have corrected this by using the natural logarithm (base $e$). While this change slightly affects the absolute values of BRI, the relative ranking and overall conclusions remain unchanged.
>
> > Q2: Introduction of Gumbel subset sampling.
>
> Thanks. To enable the selector to explore more possible combinations during the training stage, we employ a differentiable random sample strategy, namely, Gumbel-top-$k$ relaxations. This approach introduces randomness to increase the diversity of candidates while ensuring that the NLL loss gradient can be properly backpropagated for effective optimization.
>
> > Q3: BRI metric interpretation.
>
> BRI (Best Log Rank Integral) assesses the system’s ability to refine retrieval results over multiple interaction rounds. It measures the average area under the log-rank curve of all rounds. A lower BRI value indicates a more efficient and effective refinement process. We will make this clear in the next version.
>
> > Q4:  Input of Answerer in section 5.3.
>
> The T-ReID datasets only provide caption-image pairs of individuals, we turn to the multi-modal LLM to simulate the users. Specifically, we use LLaVA-OneVision-Qwen2-7B-ov as the Answerer. It provides the follow-up descriptions, looking at the image of the target person.
>
> > Q5: Ambiguous figure of qualitative results.
>
> The 4 images of candidates in the figure are the selected candidates and the rank represents the similarity ranking of the target person within the gallery **after** current round. We will improve the clarity of Figures 7 and 8 in the next version.

---

### Official Review · Reviewer_fi2Z · 2025-03-10

**Overall Recommendation:** 4

**Summary:**

This paper introduces interactive person re-identification, which refines partial text queries via dialogue interactions to better suit real-world scenarios. The proposed LLaVA-ReID generates discriminative questions using images and dialogue history, enhanced by an image selector and a looking-forward supervision strategy. Experimental results show significant improvements over existing methods.

**Claims And Evidence:**

In the motivation of this paper, the authors claims that users rarely provide a detailed and comprehensive account of the target person’s appearance, often resulting in partial and vague descriptions. This claim is well-founded, as it aligns with observations in real-world applications where initial queries are often incomplete.

**Essential References Not Discussed:**

N/A

**Experimental Designs Or Analyses:**

More analysis about visual and textual context should be provided.

**Methods And Evaluation Criteria:**

The method makes sense in the context of person re-id and the evaluation metrics are common.

**Other Comments Or Suggestions:**

N/A

**Other Strengths And Weaknesses:**

Strengths:

 It introduces an interactive approach to text-based person re-id that is applicable and user-friendly in real-world scenarios. The new benchmark is thoughtfully constructed to mirror actual questioning methods, thereby enhancing the practical relevance of the work. Moreover, by integrating traditional methods, the work reveals the task's potential to extract more detailed information about individuals, underscoring its innovative contribution to the field.

Weaknesses:

In Section 5.3, although it highlights the potential benefits of combining two retrieval models through an ensemble, how to apply the ensemble remains unclear. Additionally, the manuscript lacks clarity on how dialog is encoded within the retrieval model; it remains uncertain whether any preprocessing steps or LLM-based rewriting techniques were employed to adapt the training data.

**Questions For Authors:**

Please refer to the weaknesses.

**Relation To Broader Scientific Literature:**

Prior work in this field largely assumes that query descriptions are complete and well-structured, which is often not the case in real-world applications. In contrast, the authors introduce an interactive cross-modal retrieval technique that explicitly accounts for the inherent misalignment between dataset queries and practical user inputs, and address the candidate selection and tailored supervision for this fine-grained question generation task.

**Theoretical Claims:**

N/A

---

> ### Author Rebuttal · Authors · 2025-03-31
>
> Thank you for acknowledging the novelty of our method. We will answer the questions one by one.
>
> > Q1: Details about the ensemble.
>
> In our integration of LLaVA-ReID with the existing T-ReID framework, we use the human-annotated captions in the T-ReID datasets as the initial queries. We first encode an initial query $T$ using T-ReID model and compute the matching score between $T$ and gallery images:
> $$
> \operatorname{score}(I_i|T)=\operatorname{sim}(\phi^{base}_t(T), \phi^{base}_v(I_i)),
> $$
> where $\phi_t^{base}$ and $\phi_v^{base}$ are text and visual encoder of T-ReID model, $I_i$ is the $i$-th image in gallery, and $\operatorname{sim}(\cdot, \cdot)$ denotes cosine similarity. Next, we perform 5 rounds of interaction and encode the dialogue history $D=\{T, A_1,\dots, A_t\}$ using our Retriever, then compute the matching score:
> $$
> \text{score}(I_i|D)=\operatorname{sim}(\phi_t(D), \phi_v(I_i)).
> $$
> Then we re-rank the matching scores by simply averaging two scores:
> $$
> \text{final\\_score}_i=\frac{1}{2}(\text{score}(I_i|T)+\text{score}(I_i|D)),
> $$
> and the final score is used to compute the retrieval metrics.
>
> It’s important to note that the T-ReID dataset only provides caption-image pairs for persons. To simulate user interactions, we turn to a multi-modal LLM. Specifically, we use LLaVA-OneVision-Qwen2-7B-ov as the Answerer, which generates follow-up descriptions based on the image of the target person.
>
>
> > Q2: Preprocessing of dialogue.
>
> For dialogue preprocessing, we remove prefixes such as "Yes," "No," and multiple-choice options like "A)" and "B)" from the answers. The remaining responses are then directly concatenated and fed into the text encoder. While our current approach is straightforward, we believe that post-hoc techniques, such as dialogue rewriting, could be a promising strategy to enhance the system's ability to generalize to real-world, open-form responses.

---

### Official Review · Reviewer_RhLW · 2025-03-11

**Overall Recommendation:** 4

**Summary:**

The paper introduces a new task for person re-identification, an interactive person re-identification (I-ReID) framework that iteratively refines incomplete or vague descriptions through multi-round dialogues. The paper proposed LLaVA-ReID, a selective multi-image questioner that leverages both visual and textual contexts to generate targeted questions aimed at uncovering fine-grained differences among candidate images. The work is also supported by a newly constructed dataset which includes both coarse and fine-grained descriptions along with multi-round dialogue. Extensive experiments demonstrate its effectiveness over existing methods in both interactive and traditional text-based ReID tasks.

## update after rebuttal

After rebuttal, I have no further questions. I am inclined to maintain the score as Accept.

**Claims And Evidence:**

The paper claims that interactive refinement via multi-round dialogues can significantly improve person re-identification accuracy compared to static, text-based approaches. These claims are verified by experimental evidence on the Interactive-PEDES dataset and integration with existing T-ReID frameworks.

**Essential References Not Discussed:**

N/A

**Experimental Designs Or Analyses:**

Experimental evaluations are comprehensive.

**Methods And Evaluation Criteria:**

The proposed method for multi-round interactive retrieval is reasonable. The evaluation criteria about retrieval metrics and interaction efficiency are appropriate for assessing the system’s performance.

**Other Comments Or Suggestions:**

Please refer to the "Strengths And Weaknesses"

**Other Strengths And Weaknesses:**

Strength:
1. This manuscript presents an interactive framework for person re-identification that more closely mirrors real-world scenarios by engaging in multi-round dialogue to iteratively refine descriptions, thereby addressing a key limitation in traditional text-based ReID methods.
2. The construction of the Interactive-PEDES dataset with its detailed coarse and fine-grained descriptions and multiple dialogue rounds, providing a robust benchmark that can drive further research in the area, filling an important gap in available data.
3. The experimental evaluation is thorough, including integration with existing text-based ReID frameworks and detailed analysis of both image and question selection strategies.

Weaknesses:
1. While the candidate selection mechanism particularly the use of Gumbel-top-k relaxation is effective, the paper provides limited in-depth analysis of its robustness.
2. The computational complexity associated with processing multiple images and iterative dialogue rounds may hinder real-time application.

**Questions For Authors:**

1. Can you provide further analysis or experiments to assess the computational efficiency of your candidate selection module compared to baseline methods?
2. How do you define the parameter $k$ in the Gumbel-top-k relaxation within your selection strategy?
3. In the LLaVA-ReID framework, does the conversation get embedded into the LLaVA module before the candidate images are appended?
4. In Figure 3, could you explain the meaning of the term “clue”?

**Relation To Broader Scientific Literature:**

The work is well-situated within the fields of interactive cross-modal retrieval and text-based person re-identification. It builds upon recent approaches by addressing the fine-grained differences that traditional interactive retrieval methods have not adequately considered, thereby contributing meaningfully to the literature.

**Theoretical Claims:**

There is no theoretical claim.

---

> ### Author Rebuttal · Authors · 2025-04-01
>
> Thanks for your comments. We will answer your questions one by one in the following.
>
> > Q1: Can you provide further analysis or experiments to assess the computational efficiency of your candidate selection module compared to baseline methods? (As well as the question generation time)
>
> We conduct the experiment on the computation time of different select strategies, including top-k, k-means, and our selector. The experiment is performed on NVIDIA RTX 3090 GPUs, and we report the total selection time over five rounds in the table below:
>
> | Select Strategy | Time (second per person) | R@1/R@5   |
> | --------------- | ------------------------ | --------- |
> | selector        | 0.002195                 | 73.2/90.6 |
> | k-means         | 0.028286                 | 72.5/90.1 |
> | uniform         | 0.000099                 | 72.7/90.0 |
> | top-k           | 0.000187                 | 72.5/89.8 |
>
> Our lightweight transformer-based selector achieves the highest retrieval accuracy while ensuring no significant computational overhead compared to other methods.
>
> We also provide the inference time for 5-round question generation of LLaVA-ReID and other baseline methods are shown below:
>
> | Method         | Time (second per person) | R@1 |
> | -------------- | ------------------------ | ----------------- |
> | SimIRV         | 0.007                    | 61.3              |
> | ChatIR         | 0.216                    | 63.9              |
> | PlugIR         | 13.06                    | 65.4              |
> | **LLaVA-ReID** | **0.621**                | **73.2**          |
>
> Our multi-image questioner does not add significant computational overhead, indicating its suitability for real-time applications.
>
> > Q2: How do you define the parameter $k$ in the Gumbel-top-k relaxation within your selection strategy?
>
> The number of candidate images plays a crucial role in providing rich visual context to the Questioner. If too few candidates are given, the Questioner may not receive sufficient fine-grained information between persons. Conversely, an excessive number of images can overwhelm the MLLM’s capacity, introducing redundancy and extra computational costs. To further analyze the impact of $k$, we present retrieval results for different numbers of candidate images in the table below:
>
> | Number of Candidates | R@1       | R@5      | R@10     | mAP       | BRI       |
> | -------------------- | --------- | -------- | -------- | --------- | --------- |
> | 2                    | 72.4      | 89.7     | 94.5     | 53.2      | 0.738     |
> | **4 (ours)**         | **73.2** | **90.6** | **96.0** | **53.3** | **0.719** |
> | 6                    | 71.4      | 89.2     | 94.2     | 52.8      | 0.764     |
> | 10                   | 71.6      | 88.9     | 93.8     | 52.7      | 0.772     |
>
> The results indicate that setting $k=4$ provides the best balance between retrieval performance and computational efficiency.
>
> In our previous experiments, the reported BRI was computed using a logarithm with base 2. According to the recent released code of BRI [1], we have corrected this by using the natural logarithm (base $e$). While this change slightly affects the absolute values of BRI, the relative ranking and overall conclusions remain unchanged.
>
> > Q3: In the LLaVA-ReID framework, does the conversation get embedded into the LLaVA module before the candidate images are appended?
>
> As depicted in Equation 3, candidate images are inserted before the dialogue history, in accordance with the default configuration of LLaVA. For improved visual clarity, we modify the input order in Figure 3.
>
> > Q4: In Figure 3, could you explain the meaning of the term “clue”?
>
> The term "clue" refers to the embeddings of the context description $z_t$. We use this term to indicate that the candidates are selected based on the current "clue" we have at hand. We apologize for any confusion caused by this terminology and will revise it in the next version.
>
> [1] Lee S, Yu S, Park J, et al. Interactive Text-to-Image Retrieval with Large Language Models: A Plug-and-Play Approach[C]//Proceedings of the 62nd Annual Meeting of the Association for Computational Linguistics (Volume 1: Long Papers). 2024: 791-809.

---

> > ### Comment · Reviewer_RhLW · 2025-04-02
> >
> > Thanks for the detailed response and I have no further questions. I am inclined to maintain the score as Accept.

---

### Decision · Program_Chairs · 2025-05-01

**Decision:**

Accept (poster)

**Comment:**

This paper received four positive ratings, with all reviewers generally inclined to accept it.
The paper proposes a new task, interactive person re-identification (I-ReID), to address the problem of incomplete person description in real-world text-based person ReID. For this task, it constructs a new dataset that includes coarse and fine-grained descriptions and multiple rounds of dialogue, and introduces a question model LLaVA-ReID. All reviewers approved of the proposed I-ReID settings, believing that it more closely reflects real-world scenarios and helps improve the applicability of text-based ReID in real-world cases. They find that the constructed datasets and benchmarks were well thought out, broke through the data limitations of existing benchmarks, and promoted further research in this field. The authors provide extensive experimental evaluations and detailed analyses to demonstrate the effectiveness of the proposed LLaVA-ReID model for solving the I-ReID task.
Furthermore, all weaknesses have been addressed in the authors' response. Therefore, the Area Chair (AC) recommends accepting the paper.